# Genome-Wide Identification and Transcriptional Expression Profiles of Transcription Factor *WRKY* in Common Walnut (*Juglans regia* L.)

**DOI:** 10.3390/genes12091444

**Published:** 2021-09-19

**Authors:** Fan Hao, Ge Yang, Huijuan Zhou, Jiajun Yao, Deruilin Liu, Peng Zhao, Shuoxin Zhang

**Affiliations:** 1College of Forestry, Northwest A&F University, Xianyang 712100, China; haofan@nwafu.edu.cn (F.H.); ericguanyuzhao@163.com (H.Z.); 2College of Animal Science and Technology, Northwest A&F University, Xianyang 712100, China; geyang0125@163.com; 3Key Laboratory of Resource Biology and Biotechnology in Western China, Ministry of Education, College of Life Sciences, Northwest University, Xi’an 710069, China; yjj_zj@163.com (J.Y.); liuderuilin@163.com (D.L.); pengzhao@nwu.edu.cn (P.Z.); 4Qinling National Forest Ecosystem Research Station, Huoditang, Ankang 711600, China

**Keywords:** *Juglans regia*, *WRKY*, miR156, protein interaction

## Abstract

The transcription factor *WRKY* is widely distributed in the plant kingdom, playing a significant role in plant growth, development and response to stresses. Walnut is an economically important temperate tree species valued for both its edible nuts and high-quality wood, and its response to various stresses is an important factor that determines the quality of its fruit. However, in walnut trees themselves, information about the *WRKY* gene family remains scarce. In this paper, we perform a comprehensive study of the *WRKY* gene family in walnut. In total, we identified 103 *WRKY* genes in the common walnut that are clustered into 4 groups and distributed on 14 chromosomes. The conserved domains all contained a WRKY domain, and motif 2 was observed in most *WRKYs*, suggesting a high degree of conservation and similar functions within each subfamily. However, gene structure was significantly differentiated between different subfamilies. Synteny analysis indicates that there were 56 gene pairs in *J. regia* and *A. thaliana*, 76 in *J. regia* and *J. mandshurica*, 75 in *J. regia* and *J. microcarpa*, 76 in *J. regia* and *P. trichocarpa*, and 33 in *J. regia* and *Q. robur*, indicating that the *WRKY* gene family may come from a common ancestor. GO and KEGG enrichment analysis showed that the *WRKY* gene family was involved in resistance traits and the plant-pathogen interaction pathway. In anthracnose-resistant F26 fruits (AR) and anthracnose-susceptible F423 fruits (AS), transcriptome and qPCR analysis results showed that *JrWRKY83*, *JrWRKY73* and *JrWRKY74* were expressed significantly more highly in resistant cultivars, indicating that these three genes may be important contributors to stress resistance in walnut trees. Furthermore, we investigate how these three genes potentially target miRNAs and interact with proteins. *JrWRKY73* was target by the miR156 family, including 12 miRNAs; this miRNA family targets *WRKY* genes to enhance plant defense. *JrWRKY73* also interacted with the resistance gene *AtMPK6*, showing that it may play a crucial role in walnut defense.

## 1. Introduction

*WRKY* transcription factors are ubiquitous among higher plants, and they harbor a highly conserved WRKYGQK amino acid sequence that is followed by a zinc-finger motif at the N-terminal domain [1,2]. In important crops, *WRKY* genes have been examined in their related genomes, including 75 in peanut [3], 174 in soybean [4], 148 in *Brassica oleracea* [5], 92 in quinoa (*Chenopodium quinoa*) [6], 126 in *Raphanus sativus* [7], 79 in potato (*Solanum tuberosum*) [8], 112 in *Gossypium raimondii* and 109 in *Gossypium arboreum* WRKY [9], in cotton (116 in *G. raimondii* and 102 in *Gossypium hirsutum*) [10], 89 in rice [11], 63 in *Dendrobium officinale* [12] and 86 in Barley [13]. In woody plants, *WRKY* genes have been examined in their related genomes, namely 147 genes in *Musa acuminate* and 132 in *Musa balbisiana* [14], 58 in castor bean [2], 54 in pineapple [15], 56 in tea [16], 51 in *Citrus sinensis* [16], 48 in *Citrus clementina*, 79 in *Citrus unshiu* [17], 69 in *Dimocarpus longan* [18], 58 in moso bamboo [19], 85 in *Salix suchowensis* [20], 59 in peach [21], 95 in *Dendrobium officinale* [22], 71 in *Fragaria vesca* [23], 53 in *Elaeis guineensis* [24], 71 in sesame [25] and 53 in *Caragana intermedia* [26]. The *WRKY* gene is an important factor in the regulation of plant growth and development, as well as in a plant’s response to different kinds of stress [27], including drought, dehydration and salt stress [28]; however, the most important aspect of the *WRKY* gene is its ability to respond to abiotic stresses. This has been seen in peanut *WRKY*1 and *WRKY*12 genes, which were upregulated with salt (SA) and jasmonate (JA) treatment [28], while two abiotic stresses (salt and cold) were observed in *Raphanus sativus* in relation to heat, salinity and heavy metals [7]. In rice, *WRKY* gene family members with roles in drought tolerance and transgenic crops [29] showed a response to cold stress and methyl jasmonate (MeJA) treatments [22]. In *Brassica rapa*, *WRKY* gene family members act against abiotic and biotic stresses [30,31]. 

The common walnut (*Juglans regia* L.), i.e., the English walnut, is one of the most important hardwood trees in the world, and it is famous for its economic value, edible nuts and nutritional value [32,33,34]. Walnut oil, a high-valued oil product, is extracted from walnut kernel and used widely in food and health care industries [32,33,34]. There is no previous study regarding the *WRKY* gene in common walnut (*J. regia*) [33,34]. In a recent publication of high-quality common walnut genome data, some gene families and transcript factors were reported in common walnut, such phenylalanine ammonialyase (*PAL)*, *F-box*, fatty acid desaturase (*FAD)*, heat stress transcription factors (*HSFs*), nascent polypeptide-associated complex protein (*NAC)* and repressor of GAI; gibberellic acid-insensitive RGA and scarecrow SCR (*GRAS*) were also reported [35,36,37,38,39,40]. However, comprehensive information regarding the functional characterization of the *WRKY* gene family in common walnut is still unclear. 

In this study, based on the whole-genome sequencing of walnut, we performed a genome-wide identification of the transcription factor *WRKY* in *J. regia*. We systematically characterized *WRKY* transcription factors in common walnut. We revealed the phylogenetic tree, structural features, duplication and conserve motifs of *JrWRKYs*. To understand the expression profiles of *JrWRKYs*, we studied the transcriptional levels of *JrWRKYs* in anthracnose-resistant F26 (AR) and anthracnose-susceptible F423 (AS) fruits. Our results provide useful theoretical support for the functional characterization of these *JrWRKY* transcription factors that are involved in resistance in common walnut.

## 2. Materials and Methods

### 2.1. Bioinformatics Analysis of Putative WRKY from Walnut

An entire protein sequence of common walnuts was downloaded from NCBI (https://www.ncbi.nlm.nih.gov/genome/?term=Juglans+regia accessed on 20 December 2020) [41]. Arabidopsis *WRKY* family members were downloaded from the Arabidopsis Information Resource website (TAIR, https://www.arabidopsis.org/index.jsp accessed on 21 December 2020) using a basic local alignment search tool (BLAST) to search for prevalent walnut protein sequences, including the Arabidopsis *WRKY* sequence as a query sequence while considering those with an E value less than 1 × 10^−10^ as a typical walnut *WRKY* sequence. The Profile Hidden Markov Model (HMM) introduced in HMMER v3.2.1 (http://hmmer.org/download.html/ accessed on 28 December 2020) and Protein Family (Pfam) database (http://pfam.xfam.org/ accessed on 28 December 2020) with default parameters were used to search for prevalent walnut *WRKY* with WRKY domains. The *WRKY* sequence name and position information was acquired through BLAST with the parameters E-value < 10–15 and ID % > 50% [42]. The *WRKY* sequences were predicted on the Plant-mPLoc website to predict subcellular localization of plant proteins, including those with multiple sites [43]. The theoretical isoelectric point and molecular weight were predicted in a ProtParam tool (https://web.expasy.org/protparam/ accessed on 5 September 2021) [44].

### 2.2. Protein Alignment, Phylogenetic Analysis, Pfam Domain Detection and Chromosome Location Analysis of Walnut WRKY Genes

The complete *WRKR* sequence of walnut was aligned by MEGA7.0 (State College, PA, USA) software with default parameters [45]. Subsequently, an unrooted alignment-based phylogenetic tree was constructed with pairwise deletion of 1000 bootstrap and Poisson models with MEGA7.0 software [45,46]. The Pfam web server (http://pfam.xfam.org/ accessed on 5 January 2021) was used to identify prospective domains in each sequence. We split these sequences into 3 separate subfamilies based on specific domains discovered in these *WRKY* sequences and used TBtools [47].

### 2.3. Motif Analysis, Gene Structure and Protein Structure of Walnut WRKY Genes

Feature coordinates (exon-intron boundaries) were extracted from the GFF3 annotation files of walnut. The exon-intron structure was illustrated by TBtools [47]. Identification of patterns using various pattern alignments with the default pattern-initiated (MEME) program parameters was conducted with the maximum number of patterns set to 20, and the optimum pattern width was set to 15–20 [48]. Protein structure information was predicted by an online web server of a conserved domain (https://www.ncbi.nlm.nih.gov/Structure/bwrpsb/bwrpsb.cgi, accessed on 5 January 2021) (CDD-search) [49].

### 2.4. Synteny Analysis and Calculating Ka, Ks and Ka/Ks Values of Duplicated Gene Pairs

BLASTP was used to identify potential pairs of homologous genes across multiple genomes (Evalue < 1×10^−5^, top 5 matches) [42]. The homologous gene pairs were used to identify syntenic chains through MCScanX [50]. The detected duplicate gene pairs were detected byusing MCScanX, which included whole-genome duplication (WGD), tandem duplication, segmental and other types of gene pairs [50].

### 2.5. Plant Materials, Treatments and Collections

To analyze the expression levels of *WRKY* in common walnut, 17 transcriptome data were downloaded, including a total of 17 tissues from PJ Martínez-García et al., 2016 (PRJNA291087) [51], 10 anthracnose-resistant F26 fruits from the B26 clone (AR) and 10 anthracnose-susceptible F423 fruits from the 4–23 clone (AS) of walnut [52] (PRJNA612972) (Appendix A). Cufflinks was used to [53] to quantify these gene expression levels based on fragments per million base readings per million mapped read (FPKM) values with default parameters, and expression level was calculated using Hemi 1.0 software with default parameters [54]. The DESeq R package (1.10.1) was used to identify differential gene expression (DESeq) with an adjusted *p*-value <0.05. 

### 2.6. qRT-PCR Analysis of WRKY Genes

We then verified the *Jr**WRKY* transcript factors expression profiles in common walnut by quantitative real-time PCR (qRT-PCR) reactions in different tissues (immature fruit, pistillate flower, mature pistillate flower, embryo, somatic embryo, vegetative bud, callus exterior, catkins, hull cortex, immature hull, young hull, mature hull, hull peel, immature leaves, young leaf, mature leaves, root) and AR and AS fruits and leaves [51,52] (samples and primer see details in Appendix A). The primary specificities and associated melting curves were verified before the experiment. In each experiment, three replicates were performed. Real-time amplification responses were performed on an Applied Biosystems (USA) 7500 quick real-time PCR system. The relative concentration of expression of each gene was calculated using the 2-Ct method [55] (Appendix A).

### 2.7. miRNA Predicted in Walnut WRKY Family Genes and the Interaction Network of JrWRKY Proteins

All of the genome sequences of the common walnut *WRKY* family genes were submitted as candidate genes to predict potential miRNAs by searching against the available walnut reference of miRNA sequences using the psRNATarget Server with default parameters [56]. We visualized the interactions between the predicted miRNAs and the corresponding target walnut *WRKY* genes using Cytoscape software with default parameters [57]. Persian walnut *WRKY*matched a homologous Arabidopsis *WRKY* in the BLASTP program with an E value of 1 × 10^−5^ [58]. Regarding the Arabidopsis WRKY proteins that represented the walnut *WRKY*, 102 were uploaded to the STRING website to predict protein interactions (https://string-db.org/ accessed on 6 September 2021) [59].

## 3. Results

### 3.1. Identification and Classification of WRKY Genes

In this study, we detected a total of 103 *WRKY* genes in common walnut (Figure 1; Table 1). Based on the similar domain in which walnut *WRKY* genes were identified, the *WRKY* genes were classified into four groups, and the fourth group had the largest number of genes, including 51 members. These walnut *WRKYs* ranged in length from 281 to 760 amino acids, with a molecular weight from 0.76 Da to 0.69 Da and isoelectric points ranging from 4.97 to 9.72. Subcellular localization analysis indicated that all 103 walnut *WRKY* genes were localized in the nucleus (Figure 1; Table 1 and Appendix A).

### 3.2. Phylogenetic Tree, Motif Composition, Conserved Domain and Gene Structure of WRKY Genes

According to the phylogenetic tree and motif composition, these gene families were divided into seven subfamilies (Figure 1). According to the gene structure and conserved motif distribution, *WRKY* genes showed diverse sequence structures (Figure 2, Appendix A). In the present study, 15 conserved motifs were detected in WRKY proteins, and motif 2 was observed in most proteins as a subgroup that contained the most motifs (7), while the fewest motifs were found in subgroup 6, which contained only 3 motifs (Figure 2a,b). All *WRKY* genes contained at least one WRKY conserved domain, but subgroup 3 *WRKY* genes contained 2 WRKY domains; only two genes (*JrWRKY76* and *JrWRKY77*) contained WRKY and CCCC73 domains (Figure 2c). In addition, *WRKY* genes were diverse in terms of gene structure, where various intron-exon numbers were observed (Figure 2d), which proved the validity of the phylogenetic tree and motif composition. The structure of *WRKY* genes in common walnut has different exon-intron organizations between subfamilies (Figure 3; Appendix A). Intron numbers 2 to 6 were found in all *WRKY* genes (Figure 2d). Subgroup 1 contained 5 to 6 exons; subgroup 2 contained 3 to 4 exons; subgroup 3 contained 5 to 6 exons; subgroup 4 contained 4 to 5 exons; subgroup 5 contained 2 exons; subgroup 6 contained 3 exons; and subgroup 7 contained 3 to 4 exons (Figure 2d). 

### 3.3. Chromosome Distribution and Synteny Analysis of WRKY Genes

All *WRKY* genes could be mapped onto 12 chromosomes of walnut. Chromosome 10 contained the highest number of *WRKY* genes (10), whereas the fewest *WRKY* genes were located on chromosome 5 (1) (Figure 4). The results showed a high synteny rate within *WRKY* genes of walnut (Figure 4, Appendix A). A total of 49 showed collinear relationships between *WRKY* genes of walnut, indicating that these genes were WGD events (Figure 4, Appendix A). In total, 20 duplicated gene pairs were found in *WRKY* gene walnut genomes, including two duplicate modes: whole-genome duplication (WGD)/segmental duplication and tandem duplication (Figure 3). WGD/segmental duplications and tandem duplications were only observed in walnut *WRKY* genes (Figure 3 and Figure 4; Appendix A). Additionally, we found 22 gene pairs were under selection—17 gene pairs were under positive selection and 5 were under negative selection, indicating that these genes were under selection in evolution (Appendix A). The syntenic relationships within Juglans showed that, between *J. microcarpa* and *J. mandshurica*, we identified pairs of homologs: 71 between *J. regia* and *J. microcarpa* and 70 between *J. regia* and *J. mandshurica*, indicating that *WRKY* genes were highly conserved among the Juglans species (Figure 5; Appendix A). Multiple colinear gene pairs were found in some selected species, namely *P. trichocarpa*, *A. thaliana*, *Olea europaea* and *Quercus robur*, which inferred that the genetic copies underwent lineage-specific expansion (Figure 6; Appendix A). These findings reveal closer relationships in *J. mandshurica* species compared to other selected species, which is consistent with their evolutionary distance. Furthermore, our results imply that continuous colinear gene pairs were found in *P. trichocarpa*, *A. thaliana*, *Olea europaea* and *Quercus robur*; therefore, we suggest that the *WRKY* gene might have come from the same ancestor (Figure 3, Figure 5 and Figure 6; Appendix A). 

### 3.4. GO and KEGG Enrichment Analysis of WRKY Gene Family in Walnut

We also investigated the function annotation of the *WRKY* gene family in walnut. GO enrichment analysis showed that the top five GO terms were response to heat, pollen-pistil interaction, recognition of pollen, response to temperature stimulus and multicellular organism processes of the biological process (Figure 7a; Appendix A); however, KEGG enrichment analysis showed that the most prevalent term was the plant-pathogen interaction pathway (Figure 7b). Combining these two analytic approaches shows that the *WRKY* gene family might play an important role in a plant’s response to biotic and abiotic stresses (Figure 7; Appendix A).

### 3.5. Three Genes (JrWRKY83, JrWRKY73 and JrWRKY74) May Be Involved in Resistance Traits of Walnut, Based on Transcriptome Data and qPCR

The GO and KEGG enrichment analysis results showed that *JrWRKYs* were differently expressed in different tissues, indicating that these genes have a variable function (Figure 8a; Appendix A). In total 5 of 102 walnut *WRKY* members were expressed highly in peel compared to other tissues, particularly *JrWRKY93*, *JrWRKY94*, *JrWRKY83*, *JrWRKY73* and *JrWRKY74* (Figure 8a; Appendix A). A mean box plot shows that there were three genes (*JrWRKY83*: 546-fold, *JrWRKY73*: 307-fold and *JrWRKY74*: 1920-fold) highly expressed in anthracnose-resistant F26 fruits (including 10 replicates), while these were lowly expressed in anthracnose-susceptible F423 fruits (10 replicates) (Figure 8b; Appendix A). The morphology of leaves and fruits can be seen in the Xiangling and Shaanhe 5 cultivars, as shown in Figure 8c,d. To verify the transcriptome data, we found that there were three genes that were highly expressed in the Xiangling cultivar. We performed a qPCR analysis to verify these results. Based on our real-time PCR results, we observed that *JrWRKY83* (23-fold), *JrWRKY73* (10-fold) and *JrWRKY74* (11-fold) were highly expressed in resistance traits compared to non-resistance traits, including leaf and fruit (Figure 8e,f; Appendix A). 

### 3.6. MicroRNA Targeting and WRKY Interaction Network 

To understand the underlying regulatory mechanism of miRNAs involved in the regulation of *WRKYs*, we identified 206 putative miRNAs targeting 45 common walnut *WRKY* genes (Figure 9a; Appendix A). The most target genes were *JrWRKY65* and *JrWRKY67*, containing 197 miRNAs, while the least targeted gene was *JrWRKY55*, containing 62 miRNAs (Figure 9a; Appendix A). Based on transcriptome profile and qPCR results, we selected *JrWRKY73* and the related 85 miRNAs to construct a relationship network using Cytoscape software (Figure 9a; Appendix A). Of these 85 miRNAs, we found that the miRNA family with the closest relationship was *JrWRKY73*, which was targeted by the Jre-miR156 family, including 12 miRNAs (Jre-miR156a, Jre-miR156b, Jre-miR156c, Jre-miR156d, Jre-miR156e, Jre-miR156f, Jre-miR156g, Jre-miR156h, Jre-miR156i, Jre-miR156j, Jre-miR156k, Jre-miR156l) (Figure 9a; Appendix A). Each *JrWRKYs*was in close association with at least one WRKY protein from Arabidopsis. Some *JrWRKYs* proteins were closely aligned with the same WRKY protein in Arabidopsis. We downloaded *WRKYs* from Arabidopsis to detect the predicted role of highly expressed genes in the fruits and leaves of AR of Persian walnut. A previous study claimed that these genes regulate the development of fruits and are responsible for stress. Therefore, we detected the interaction relationship between these genes, and the results indicate a strong relationship between *JrWRKY73s* and *AtCYP78A9*, *AtMPK6*, *AtMPK10*, *AtARF19* and *AtCYP78A9* (Figure 9b).

## 4. Discussions

### 4.1. The Gene Family Member among Plants

*WRKY* plays a critical role in plant growth, development and resistance. Recently, there have been reports on the functional analysis of *WRKY* genes in plants [3,7,8,10,22,30,31]. However, their complex polyploidy and lack of genomic information have limited further study. *WRKY* is a large gene family in the plant kingdom, and the number of genes in the family varies from 48 to 148. This study demonstrated that the *WRKY* gene family contained 103 members in walnut. Comparative analysis showed that the number of *WRKY* genes in each plant was not determined the genome size of each plant; for instance, the maize genome was about 2300 Mb, the Arabidopsis genome was about 125 Mb and the rice genome was about 389 Mb, while the common walnut was about 584 Mb. As such, genome size was not the main determiner of the number of gene families [60,61] (Figure 1, Figure 2 and Figure 4). The *WRKY* gene family in walnut may also have a basic capacity to resist stress from cold, salt and disease [31]. Recently, some studies have shown that the *WRKY* gene is localized to the nucleus [62,63]. Our results also show that all 103 *WRKY* genes were predicted to be in the nucleus of common walnut (Table 1). 

### 4.2. The Evolution of WRKY Gene Family in Walnut

The *WRKY* gene family in walnut can be divided into four groups, similar to the classification of *WRKY* genes in *Musa acuminate* and *Musa balbisiana*, Castor bean, pineapple, soybean, *C. sinensis*, *C. clementina* and *C. unshiu*, *Eucalyptus grandis*, Quinoa, *Dimocarpus longan*, *Raphanus sativus*, potato, moso bamboo, *G. raimondii* and *G. arboretum*, Cassava, willow, *Oryza officinalis*, peach and *Dendrobium officinale*. The division of the family in such a manner suggests that the results of our classification were reasonable and reliable [25,26,27,28,29,30,31,32,33,34,35,36,37,38,39,40]. In view of the conserved motifs, motifs 5, 9, 1, 2, 8, 10 and 7 were the typical motifs of group I; motifs 1 and 2 were the typical motifs of group II; and motifs 4, 1 and 2 were the typical motifs of group III. This result is consistent with our pervious study (Figure 2) [2]. Despite strong conservation of their DNA-binding domain, the overall structures of *WRKYs* are highly divergent and can be categorized into distinct groups, which reflects their different functions [62]. The intron and exon structures of a gene family always provide clues that demonstrate its evolution [63]. The coding sequence position and length of a single gene are always determined by intron positions, which contribute to the protein diversity caused evolution [63,64,65,66]. Duplication modes of genes, such as WGD or segmental duplication, tandem duplication and dispersed duplication, are characteristic features of the evolution of eukaryotic genomes [67]. Tandem and segmental duplication events played a critical role in the expansion of the *WRKY* gene family [8,9,24,28,68]. Whole-genome duplication events are common during angiosperm evolution and usually lead to the expansion of gene families [5,60]. In walnut, we found no tandem or segmental duplication events. This strongly indicates that the *WRKY* gene family members of walnut are predominantly influenced by WGD events, a finding that is consistent with previous studies [8,9,24,28,60,61] (Figure 3, Figure 5 and Figure 6).

### 4.3. The Function of WRKY Gene Family 

The *WRKY* gene family is involved in many important biology processes, though the most important are response to abiotic stresses, pathogen defense, senescence and trichome development [3,7,8,10,22,30,31,63]. There were 20% (4/20) GO biological process pathways and 50% (1/2) KEGG pathways that were enriched in resistance pathways, which is consistent with previous studies claiming that *WRKY* genes work against abiotic and biotic stresses [30,31]. We were interested in *WRKY* genes that regulate the development of resistance traits; therefore, we analyzed public transcriptome data for *J. regia* (Appendix A, Figure 8) and discovered that many family members were highly expressed in the hull. The hull is always impacted by stress [52,69], indicating that *WRKYs* may be responsible for conferring stress resistance in walnut (Figure 8) [30,31]. With the transcriptome data of AR fruits and AS fruits, we investigated the expression profile between these two fruits; three genes, *JrWRKY83*, *JrWRKY732* and *JrWRKY74*, were highly expressed in AR fruits, indicating that these three genes increased their expression level when infected by the stress of *Colletotrichum gloeosporioides*, which predominantly affects walnut anthracnose through *C. gloeosporioides* can cause leaf scorches or defoliation, as well as fruit gangrene, which is currently the most challenging disease in walnut production [52,69,70]. In line with previous studies, we collected leaves and fruits infected by *C. gloeosporioides*. Regarding our real-time PCR results, *JrWRKY83*, *JrWRKY732* and *JrWRKY74* were induced by *C. gloeosporioides* stress in the leaves and root tissues of walnut cultivars. *JrWRKY83*, *JrWRKY732* and *JrWRKY74* were more upregulated in response to *C. gloeosporioides* stress. In recent years, many studies have shown that miRNAs in plants respond primarily to stress by regulating the expression of genes associated with stress [71]. In terms of *WRKY*, some researchers have reported that Md-miR156ab and Md-miR395 resulted in a significant reduction in *MdWRKYN1* and *MdWRKY26* expression [72]. *HaWRKY6* is a particularly divergent *WRKY* gene exhibiting a putative target site for the miR396; thus, the possible post-transcriptional regulation of *HaWRKY6* by miR396 was investigated [73]. In our study, we found that 12 miRNAs of the miRNA156 family targeted the potentially resistant gene *JrWRKY73*, and we also reported that Md-miR156ab targeted *WRKY* transcription factors to influence apple resistance to leaf spot disease [72]. The diverse patterns of microRNA targeting *WRKY* genes indicate that the networks of microRNA156 and JrWRKY73 may be key regulator networks for the *WRKY* gene family in common walnut. The results of the interaction indicate a strong relationship between *JrWRKY73* and *AtCYP78A9* [74], *AtMPK6* [75], *AtMPK10* [76] and *AtARF19* [77]. *AtCYP78A9* induces large and seedless fruit in Arabidopsis, indicating that *JrWRKY73* may participate in the development of walnut fruits [74]. *JrWRKY73* interacts with *AtMPK10* and *AtMPK13*, while *AtMKK6* and *AtMPK4* activate *AtMPK13* and interact with *AtMPK12* in yeast cells, indicating that they may have the same function [76]. *JrWRKY73* interacts with the activator of a cholera toxin known as *AtARF19*, indicating that *JrWRKY73* may have the same function [77]. *JrWRKY73* showed a higher expression level in AS fruits when induced by *C. gloeosporioides* stress, which was consistent with previous studies reporting that *ATMPK6* was involved in distinct signal transduction pathways responding to these environmental stresses [75]. These protein interactions showed that *JrWRKY73* may play a key role in fruit development, yeast cells, activation of a cholera toxin and resistance in walnut. However, when combined with the expression profile, miRNA-targeted network and protein interacted network, the results showed that *JrWRKY73* played critical role in walnut defense. Moreover, these findings could lay a theoretical foundation for the functional study of *JrWRKYs* and the further construction of common walnut resistance regulation networks.

## 5. Conclusions

In this study, we identified 103 *WRKY* genes in walnut. Phylogenetic analysis showed that the *WRKY* genes could be grouped into four groups (Figure 1). These walnut *WRKY* genes are distributed on 16 different chromosomes (Figure 2). A phylogenetic analysis and synteny analysis showed that this gene family was conserved in evolution (Figure 3, Figure 5 and Figure 6). Tissue expression profiles of the *WRKY* genes demonstrated that the *WRKY* gene family might play a vital role in resistance traits (Figure 8). Three genes (*JrWRKY83*, *JrWRKY73* and *JrWRKY74*) were highly expressed in resistant cultivars compared to susceptible varieties (Figure 8). Furthermore, 206 putative miRNAs targeting 45 common walnut *WRKY* genes, especially *JrWRKY73*, were targeted by the Jre-miR156 family, including 12 miRNAs, and it was reported that this miRNA family could target WRKY genes to enhance disease resistance in plants. *JrWRKY73* interact with four genes *AtCYP78A9*, *AtMPK6*, *AtMPK10* and *AtARF19*, indicating that *JrWRKY73* plays a crucial role in plant defense (Figure 9).

## Figures and Tables

**Figure 1 genes-12-01444-f001:**
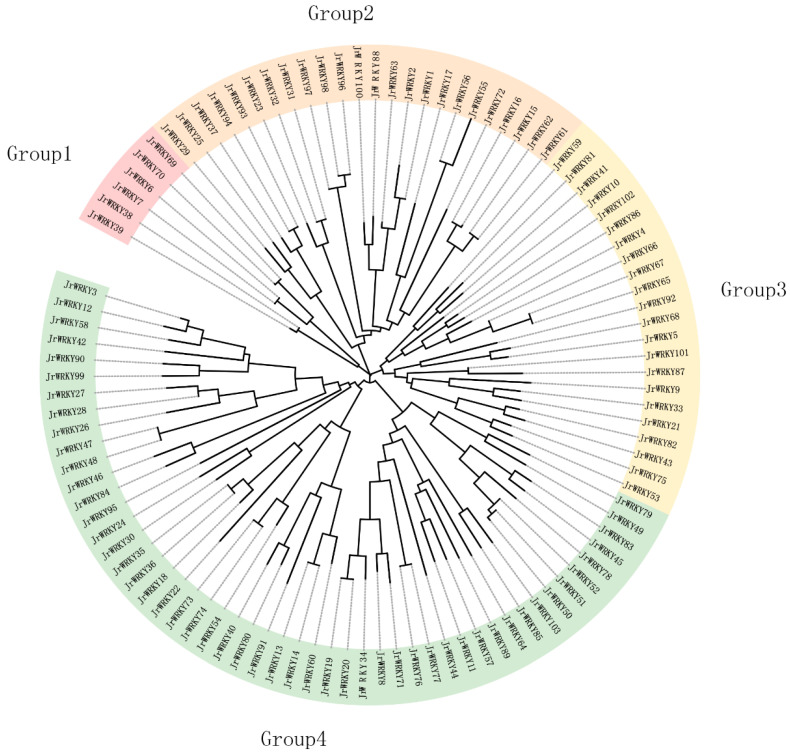
The phylogenetic tree of *WRKY* gene family in walnut based on the latest genome (NCBI version: GCA_001411555.2 Walnut 2.0). These 103 sequences were used to construct a neighbor-joining (NJ) tree. The tree was divided into four subfamilies; the names of different groups are displayed.

**Figure 2 genes-12-01444-f002:**
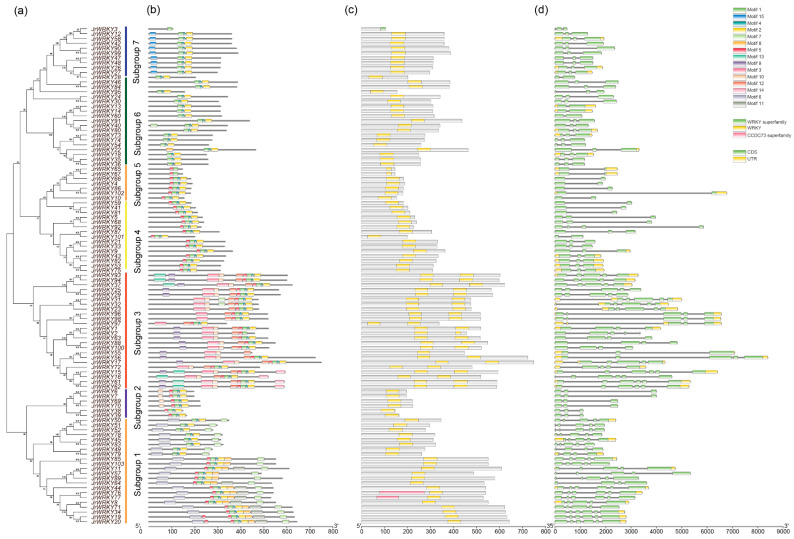
Phylogenetic analyses of the motifs in proteins and gene structures of *WRKY* genes, (**a**) phylogenetic tree, (**b**) conserved motif distribution of *WRKY* genes, (**c**) the conserved domains and (**d**) intron-exon distribution of *WRKY* genes.

**Figure 3 genes-12-01444-f003:**
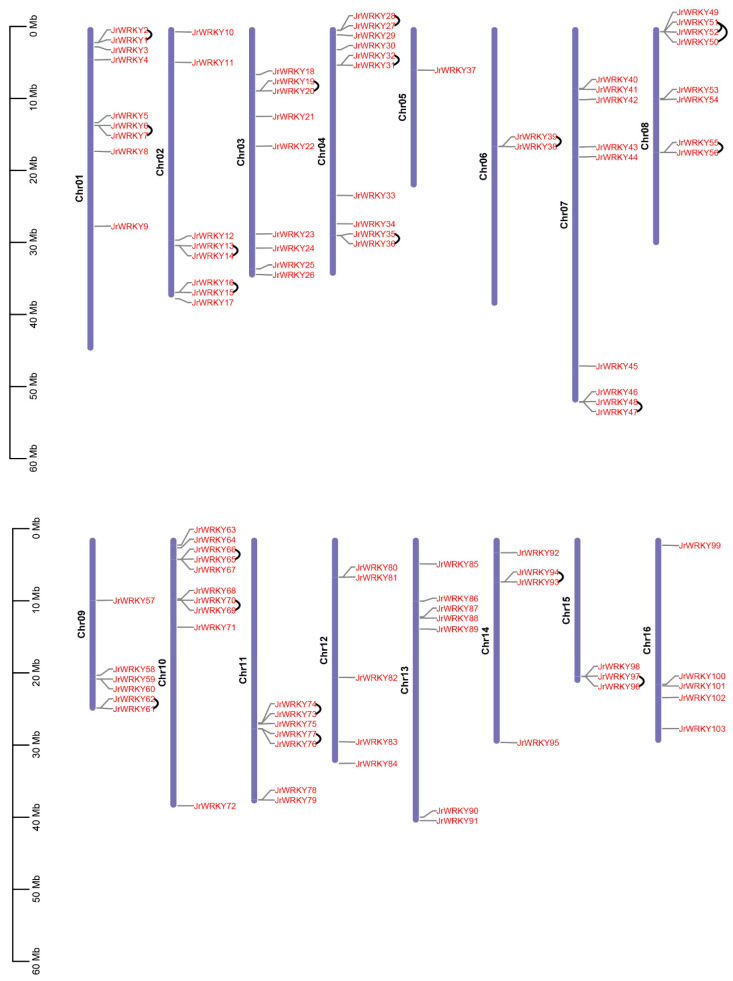
Chromosome location and tandem analysis of *WRKY* genes. The blue boxes represent chromosomes of walnut; black lines represent the tandem relationships of *WRKY* genes in walnut.

**Figure 4 genes-12-01444-f004:**
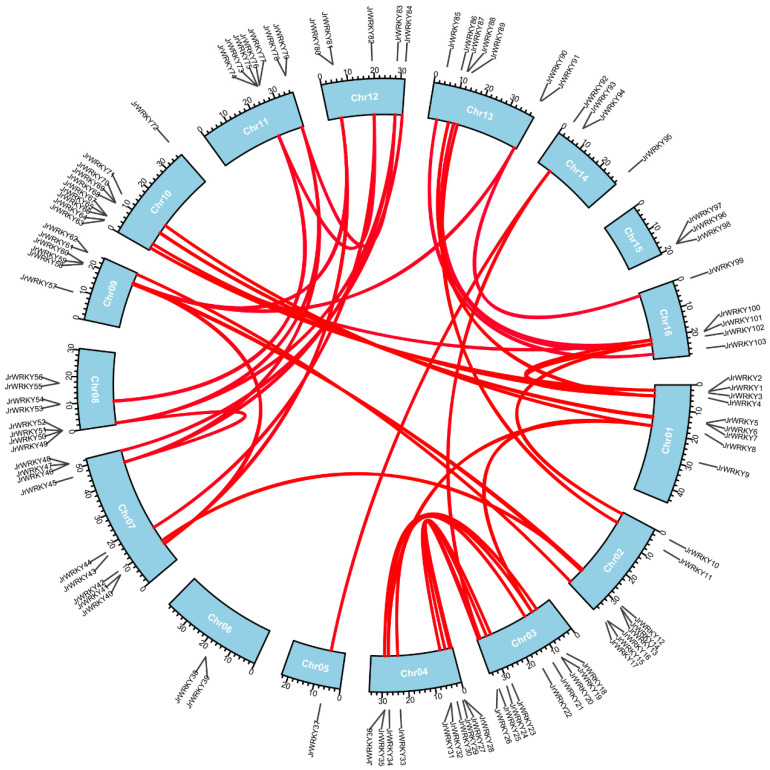
Chromosome location and synteny analysis of *WRKY* genes within walnut genome. The blue boxes represent chromosomes of walnut; red lines represent the syntenic relationships of *WRKY* genes in walnut.

**Figure 5 genes-12-01444-f005:**
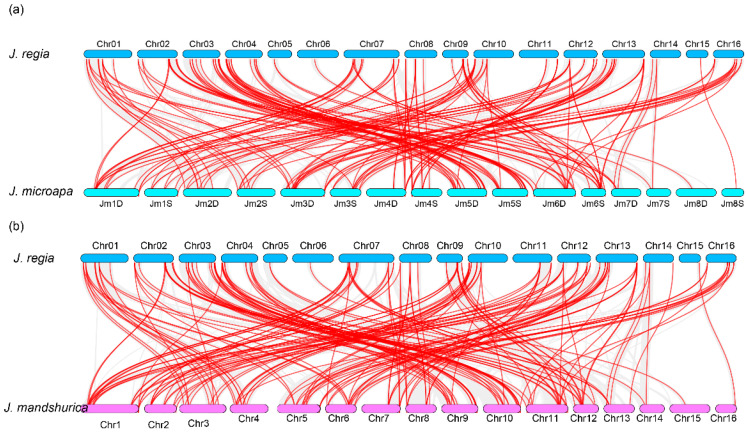
Syntenic analysis of *WRKY* genes in Juglans species, including *J. microcarpa* and *J. mandshurica*. (**a**) the syntenic analysis of *WRKY* genes between *J. regia* and *J. microcarpa*. (**b**) the syntenic analysis of *WRKY* genes between *J. regia* and *J. mandshurica*.

**Figure 6 genes-12-01444-f006:**
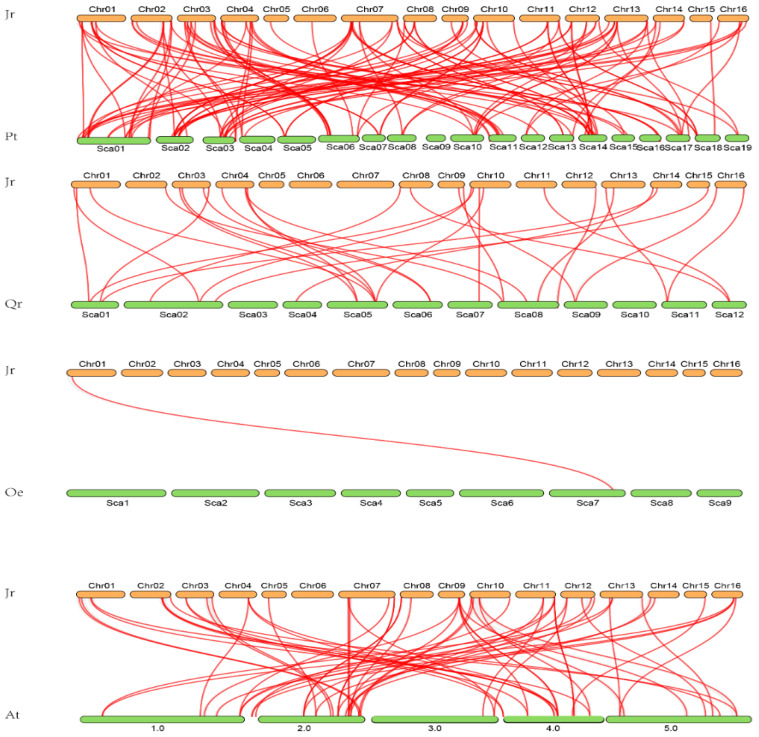
Syntenic analysis of *WRKY* genes with other species. Pt indicates *P. trichocarpa*; At indicates *A. thaliana*; Oe indicates *Olea europaea*; Qr indicates *Quercus robur*.

**Figure 7 genes-12-01444-f007:**
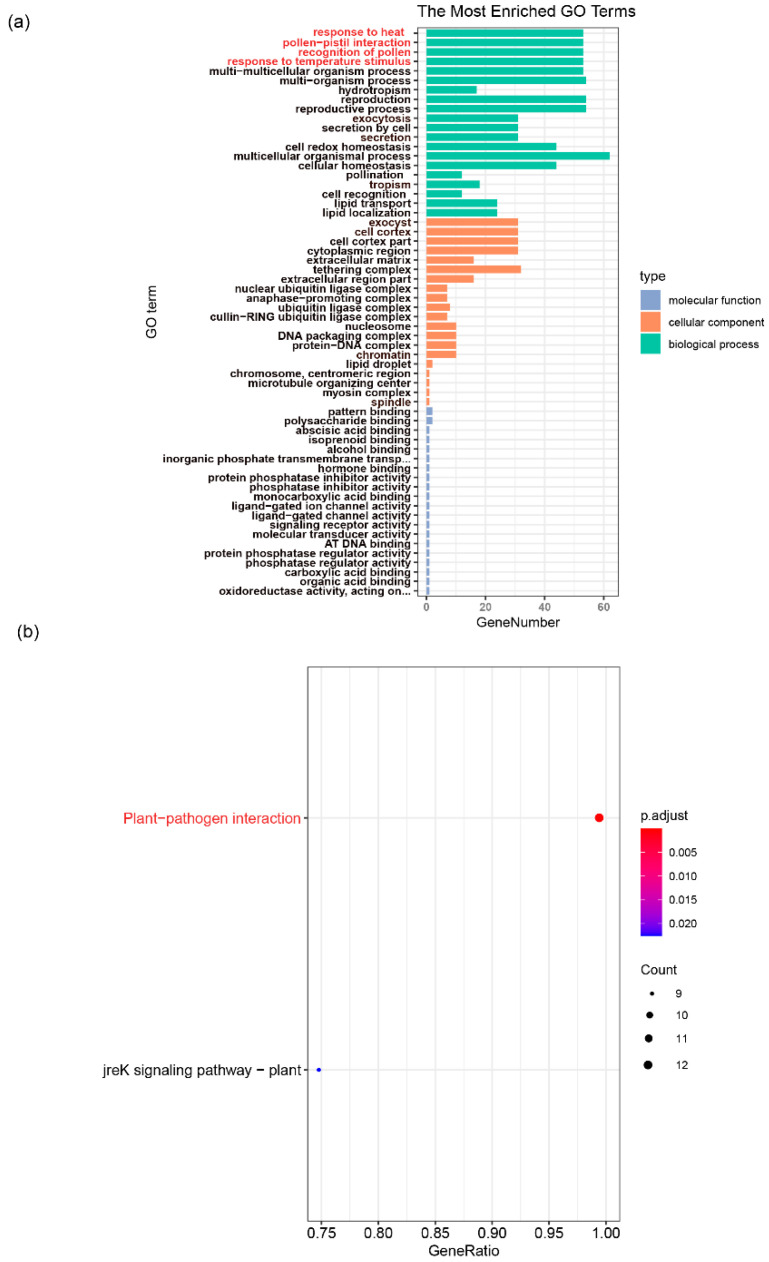
GO and KEGG enrichment analysis of *WRKY* gene family in walnut. (**a**) the GO enrichment analysis of *WRKY* genes in walnut. (**b**) the KEGG enrichment analysis of *WRKY* genes in walnut. The red marked indicates that the concerned terms in this study.

**Figure 8 genes-12-01444-f008:**
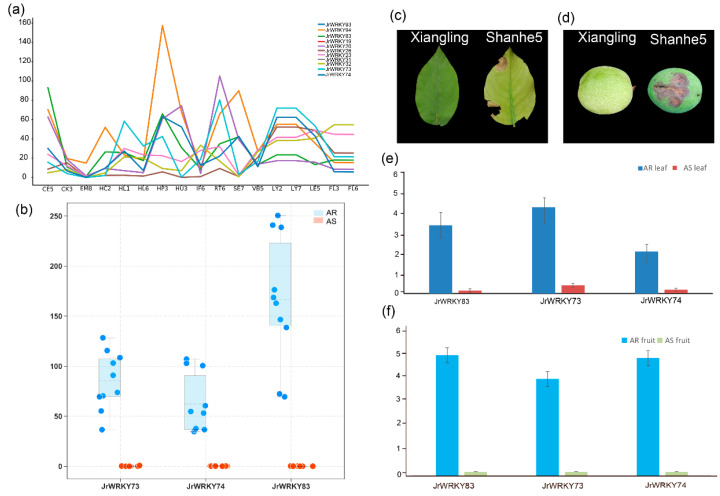
The expression profile of *WRKY* gene family in walnut: (**a**) the highly expressed genes among different tissues of walnut; IF6: immature fruit; FL3: pistillate flower; FL6: mature pistillate flower; EM8: embryo; SE7: somatic embryo; VB5: vegetative bud; CE5: callus exterior; CK3: catkins; HC2: hull cortex; HU3: immature hull; HL1: immature hull; HL6: young hull; HP3: hull peel; LY2: immature leaf; LY7: young leaf; LE5: mature leaves; RT6: root. (**b**) Mean box plot of *WRKY* members between anthracnose-resistant F26 fruits (AR) and anthracnose-susceptible F423 fruits (AS); each black circle represents each sample. (**c**,**d**) The morphology of walnut leaf (resistance (Cultivar Xiangling) and non-resistance (Cultivar Shanhe5)) and fruit (resistance (Cultivar Xiangling) and non-resistance (Cultivar Shanhe5)). (**e**,**f**) Relative expression levels of WRKY genes in walnut leaf (resistant cultivar Xiangling and non-resistant cultivar Shanhe5, and fruit resistant cultivar Xiangling and non-resistant cultivar Shanhe5).

**Figure 9 genes-12-01444-f009:**
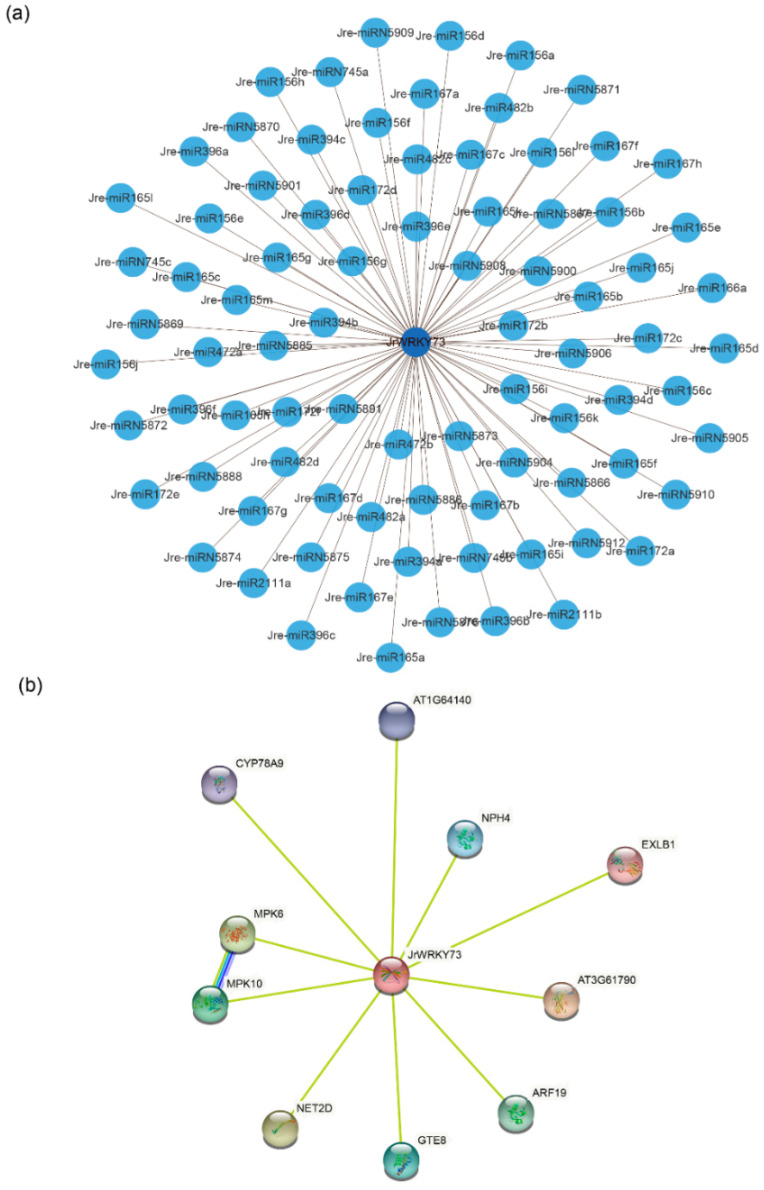
(**a**) A schematic representation of the regulatory network relationships between the putative miRNAs and their targeted walnut *WRKY* genes. (**b**) *JrWRKYs* interaction network. *JrWRKYs* interaction network was constructed using Arabidopsis homologous *WRKYs*. Proteins are represented by network nodes. The 3D protein structure is displayed inside the nodes. Edges represent associations of proteins.

**Table 1 genes-12-01444-t001:** Information on the *WRKY* gene family in common walnut.

Protein ID	Gene Name	Chr	Start	End	Subcellular Location	Molecular Weight (Da)	Theoretical pI
Jr01_03140_p1	*JrWRKY1*	Chr01	2257771	2258097	Nucleus	56,541.73	8.09
Jr01_03150_p1	*JrWRKY2*	Chr01	2257500	2257520	Nucleus	49,890.37	6.37
Jr01_03780_p1	*JrWRKY3*	Chr01	2846190	2846322	Nucleus	12,027.76	8.89
Jr01_06810_p1	*JrWRKY4*	Chr01	4635310	4635698	Nucleus	21,804.48	9.57
Jr01_15550_p1	*JrWRKY5*	Chr01	13361540	13361835	Nucleus	26,093.39	8.57
Jr01_15790_p1	*JrWRKY6*	Chr01	13757222	13757429	Nucleus	21,922.28	6.52
Jr01_15800_p1	*JrWRKY7*	Chr01	13757222	13757429	Nucleus	22,078.47	6.95
Jr01_18750_p1	*JrWRKY8*	Chr01	17355967	17356096	Nucleus	59,268.47	5.68
Jr01_22930_p1	*JrWRKY9*	Chr01	27759678	27759839	Nucleus	41,188.18	8.45
Jr02_00640_p1	*JrWRKY10*	Chr02	765843	766023	Nucleus	17,790.98	9.72
Jr02_04990_p1	*JrWRKY11*	Chr02	4986379	4986534	Nucleus	65,659.19	6.77
Jr02_16420_p1	*JrWRKY12*	Chr02	29672099	29672723	Nucleus	40,484.07	5.53
Jr02_17180_p1	*JrWRKY13*	Chr02	30435407	30435516	Nucleus	35,003.51	5.69
Jr02_17190_p1	*JrWRKY14*	Chr02	30435407	30435516	Nucleus	34,916.43	5.69
Jr02_25510_p1	*JrWRKY15*	Chr02	36928719	36929192	Nucleus	64,334.26	7.65
Jr02_25520_p1	*JrWRKY16*	Chr02	36928211	36928718	Nucleus	56,824.61	6.88
Jr02_26760_p1	*JrWRKY17*	Chr02	37818134	37818221	Nucleus	80,308.26	5.73
Jr03_08810_p1	*JrWRKY18*	Chr03	6687954	6688183	Nucleus	27,753.91	5.29
Jr03_11540_p1	*JrWRKY19*	Chr03	8968628	8969068	Nucleus	68,711.71	6.71
Jr03_11550_p1	*JrWRKY20*	Chr03	8968628	8969104	Nucleus	69,981.2	7.41
Jr03_15160_p1	*JrWRKY21*	Chr03	12506601	12506994	Nucleus	36,299.27	6.66
Jr03_17800_p1	*JrWRKY22*	Chr03	16643403	16644049	Nucleus	49,731.13	5.09
Jr03_22520_p1	*JrWRKY23*	Chr03	28822942	28823420	Nucleus	51,969.62	8.97
Jr03_23890_p1	*JrWRKY24*	Chr03	30792338	30792633	Nucleus	38,321.41	5.58
Jr03_26460_p1	*JrWRKY25*	Chr03	33675277	33675552	Nucleus	63,267.42	8.37
Jr03_27180_p1	*JrWRKY26*	Chr03	34462035	34462179	Nucleus	35,255.41	6.05
Jr04_00630_p1	*JrWRKY27*	Chr04	535739	535935	Nucleus	33,737.69	6.05
Jr04_00640_p1	*JrWRKY28*	Chr04	535288	535738	Nucleus	23,556.39	6.47
Jr04_01100_p1	*JrWRKY29*	Chr04	1179865	1180318	Nucleus	63,547.6	6.45
Jr04_02750_p1	*JrWRKY30*	Chr04	3227671	3228181	Nucleus	34,373.42	5.01
Jr04_03880_p1	*JrWRKY31*	Chr04	5390370	5390567	Nucleus	51,825.9	9.37
Jr04_03890_p1	*JrWRKY32*	Chr04	5390036	5390075	Nucleus	51,825.9	9.37
Jr04_10710_p1	*JrWRKY33*	Chr04	23465258	23465703	Nucleus	36,869.57	6.5
Jr04_13770_p1	*JrWRKY34*	Chr04	27404979	27405225	Nucleus	67,672.3	6.24
Jr04_15560_p1	*JrWRKY35*	Chr04	29034659	29034819	Nucleus	28,695.91	5.15
Jr04_15570_p1	*JrWRKY36*	Chr04	29034659	29034819	Nucleus	23,041.43	9.4
Jr05_06670_p1	*JrWRKY37*	Chr05	6064023	6064340	Nucleus	28,782.99	5.15
Jr06_10190_p1	*JrWRKY38*	Chr06	16660192	16660378	Nucleus	17,033.67	5.12
Jr06_10200_p1	*JrWRKY39*	Chr06	16660141	16660378	Nucleus	18,792.40	5.07
Jr07_07490_p1	*JrWRKY40*	Chr07	8624036	8624525	Nucleus	37,173.78	5.45
Jr07_07550_p1	*JrWRKY41*	Chr07	8732786	8732963	Nucleus	23,041.43	9.4
Jr07_08410_p1	*JrWRKY42*	Chr07	10169096	10169403	Nucleus	40,140.44	5.71
Jr07_11780_p1	*JrWRKY43*	Chr07	16745738	16745973	Nucleus	37,411.32	5.68
Jr07_12950_p1	*JrWRKY44*	Chr07	18127055	18127153	Nucleus	59,910.24	6.89
Jr07_32310_p1	*JrWRKY45*	Chr07	47146687	47146990	Nucleus	34,545.66	8.75
Jr07_38970_p1	*JrWRKY46*	Chr07	52124789	52125284	Nucleus	42,522.33	5.96
Jr07_38980_p1	*JrWRKY47*	Chr07	52129707	52130044	Nucleus	35,255.13	5.36
Jr07_38990_p1	*JrWRKY48*	Chr07	52129707	52130044	Nucleus	35,168.05	5.36
Jr08_00860_p1	*JrWRKY49*	Chr08	742451	742489	Nucleus	30,859.84	8.4
Jr08_00870_p1	*JrWRKY50*	Chr08	753017	753211	Nucleus	39,081.65	6.87
Jr08_00880_p1	*JrWRKY51*	Chr08	753017	753037	Nucleus	33,228.87	7.11
Jr08_00890_p1	*JrWRKY52*	Chr08	753017	753037	Nucleus	31,222.71	7.63
Jr08_11260_p1	*JrWRKY53*	Chr08	10002118	10002312	Nucleus	34,664.94	8.7
Jr08_11410_p1	*JrWRKY54*	Chr08	10151261	10151340	Nucleus	29,684.2	6.08
Jr08_14560_p1	*JrWRKY55*	Chr08	17508557	17508602	Nucleus	48,352.44	4.97
Jr08_14570_p1	*JrWRKY56*	Chr08	17509732	17509923	Nucleus	78,751.49	5.48
Jr09_02120_p1	*JrWRKY57*	Chr09	8767440	8767490	Nucleus	54,475.98	9.02
Jr09_09980_p1	*JrWRKY58*	Chr09	19148726	19148857	Nucleus	40,245.48	5.64
Jr09_10570_p1	*JrWRKY59*	Chr09	19658833	19659206	Nucleus	20,577.38	9.51
Jr09_10580_p1	*JrWRKY60*	Chr09	19665154	19665485	Nucleus	35,303.85	5.81
Jr09_16000_p1	*JrWRKY61*	Chr09	23697163	23697517	Nucleus	63,785.54	7.13
Jr09_16010_p1	*JrWRKY62*	Chr09	23697163	23697516	Nucleus	63,698.46	7.13
Jr10_01810_p1	*JrWRKY63*	Chr10	1104291	1104561	Nucleus	56,392.69	8.07
Jr10_02340_p1	*JrWRKY64*	Chr10	1484598	1484699	Nucleus	58,693.61	6.74
Jr10_04760_p1	*JrWRKY65*	Chr10	3087251	3087411	Nucleus	16,281.59	9.68
Jr10_04770_p1	*JrWRKY66*	Chr10	3086865	3087250	Nucleus	20,646.35	9.24
Jr10_04790_p1	*JrWRKY67*	Chr10	3099966	3100126	Nucleus	16,281.59	9.68
Jr10_10930_p1	*JrWRKY68*	Chr10	8553588	8553895	Nucleus	27,001.35	9.41
Jr10_11120_p1	*JrWRKY69*	Chr10	8740734	8740998	Nucleus	24,938.48	7.2
Jr10_11130_p1	*JrWRKY70*	Chr10	8740734	8740998	Nucleus	25,197.8	7.71
Jr10_13870_p1	*JrWRKY71*	Chr10	12479682	12480398	Nucleus	67,557.19	6.53
Jr10_25600_p1	*JrWRKY72*	Chr10	37241492	37241717	Nucleus	52,902.21	5.65
Jr11_16130_p1	*JrWRKY73*	Chr11	25746728	25746820	Nucleus	31,410.1	5.4
Jr11_16150_p1	*JrWRKY74*	Chr11	25746160	25746727	Nucleus	31,219.88	5.4
Jr11_16330_p1	*JrWRKY75*	Chr11	25890137	25890461	Nucleus	34,342.42	6.67
Jr11_17180_p1	*JrWRKY76*	Chr11	26573550	26573846	Nucleus	59,310.68	8.76
Jr11_17190_p1	*JrWRKY77*	Chr11	26573550	26573807	Nucleus	58,058.19	8.62
Jr11_30170_p1	*JrWRKY78*	Chr11	36401163	36401522	Nucleus	35,567.93	7.58
Jr11_30180_p1	*JrWRKY79*	Chr11	36421169	36421419	Nucleus	29,474.94	8.99
Jr12_04410_p1	*JrWRKY80*	Chr12	5522245	5522483	Nucleus	36,853.45	5.3
Jr12_04430_p1	*JrWRKY81*	Chr12	5552443	5552620	Nucleus	24,150.48	8.82
Jr12_10150_p1	*JrWRKY82*	Chr12	19449214	19449375	Nucleus	36,367.45	6.67
Jr12_20670_p1	*JrWRKY83*	Chr12	28364827	28365189	Nucleus	35,713.8	8.87
Jr12_25170_p1	*JrWRKY84*	Chr12	31374462	31374933	Nucleus	42,430.39	6.48
Jr13_05050_p1	*JrWRKY85*	Chr13	3716845	3717165	Nucleus	60,666.51	6.13
Jr13_12290_p1	*JrWRKY86*	Chr13	8857199	8857376	Nucleus	20,751.24	9.18
Jr13_14610_p1	*JrWRKY87*	Chr13	11027973	11028382	Nucleus	33,873.06	5.86
Jr13_14850_p1	*JrWRKY88*	Chr13	11227961	11228243	Nucleus	59,330.49	7.29
Jr13_16070_p1	*JrWRKY89*	Chr13	12734563	12734607	Nucleus	63,019.02	8.13
Jr13_30130_p1	*JrWRKY90*	Chr13	38812568	38813261	Nucleus	42,829.33	5.52
Jr13_30630_p1	*JrWRKY91*	Chr13	39302382	39302974	Nucleus	47,896.2	5.55
Jr14_02960_p1	*JrWRKY92*	Chr14	2174790	2174952	Nucleus	25,660.05	6.99
Jr14_08090_p1	*JrWRKY93*	Chr14	6185107	6185468	Nucleus	66,068.98	7.37
Jr14_08100_p1	*JrWRKY94*	Chr14	6185107	6185461	Nucleus	65,679.57	7.37
Jr14_22100_p1	*JrWRKY95*	Chr14	28465418	28465452	Nucleus	17,747.01	6.51
Jr15_12650_p1	*JrWRKY96*	Chr15	19303136	19303172	Nucleus	56,662.56	5.47
Jr15_12660_p1	*JrWRKY97*	Chr15	19302848	19303172	Nucleus	37,155	9.31
Jr15_12670_p1	*JrWRKY98*	Chr15	19303136	19303172	Nucleus	56,790.69	5.47
Jr16_00890_p1	*JrWRKY99*	Chr16	1133244	1133569	Nucleus	43,423.97	5.82
Jr16_12390_p1	*JrWRKY100*	Chr16	20419764	20420258	Nucleus	56,611.54	6.41
Jr16_12590_p1	*JrWRKY101*	Chr16	20622773	20623229	Nucleus	22,661.49	9.54
Jr16_14290_p1	*JrWRKY102*	Chr16	22251504	22251576	Nucleus	20,667.06	9.3
Jr16_19690_p1	*JrWRKY103*	Chr16	26545962	26546702	Nucleus	61,636.63	6.06

Note: protein ID, gene ID and CDS (coding sequence) ID indicate that the accession numbers of the *WRKY* gene family member sequences were downloaded from the National Center for Biotechnology (NCBI). Da indicates Daltons (unified atomic mass unit); pI indicates isoelectric point.

## Data Availability

Data sharing not applicable.

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
