# Peer review of "Genome-Wide Identification and Transcriptional Expression Profiles of Transcription Factor WRKY in Common Walnut (Juglans regia L.)"

_genes, 2021, doi:10.3390/genes12091444_

Round 1

Reviewer 1 Report

In the study, the authors performed a genome-wide study to identify the WRKY genes in common walnut. This is the first report in walnut, and the WRKY family has been widely reported in several plant species. However, this manuscript was poorly arranged, written, and completed. The authors fail to provide the main supporting results of all the analysis in the supplementary files, making it hard to evaluate the results from the figures completely. Thus, I strongly suggest read some good papers and improve your presentation accordingly. Mainly, below mentioned paper could help you to improve your work. Thus, I cannot support the paper in its present form unless it is thoroughly revised and improved. Some of the major issues are:

  1. Line 17-18, rephrase the sentence like “In total, we identified 103 WRKY genes in common walnut which are clustered into four groups and distributed on 14 chromosomes.
  2. Line 19, The authors should present the main results of the conserved domain, gene structure, and motif composition in the abstract.
  3. Update the keywords; they should not be similar to the title.
  4. Line 34-43, please rephrase the sentences for more clarity. Like, 10 members in cotton, 5 members in ABC, etc. Further, add references after every study.
  5. Line 47, 50, define SA, JA, MeJA abbreviations. Also, check the entire text for undefined abbreviations like line 66.
  6. Line 72, uploaded?? What?? Would you please check and correct the statement?
  7. Line 77, 1e10-10, correct it.
  8. Line 88, fix the spacing error.
  9. In the methodology, the authors use “WE” several times. Please avoid using we, our, etc.
  10. Line 94, it should be GSDs. Would you please rephrase the exact database name?
  11. Line 95-96, please rephrase the sentences for more clarity.
  12. Line 102, cite??? What?
  13. Line 108, add the accession number or URL which is mentioned on line 110.
  14. Line 114, change genes to Genes.
  15. Line 118, mention the tissues/organs names.
  16. Line 123, fix the superscript issue.
  17. Line 126-129, Table S3 should be included in the main text. Further, more information on the physico-chemical properties of molecular weight, isoelectric points, etc., must be added to this table. Also, explain the method for physioco-chemical properties in the methodology section. Would you please check some recently published article and improve this part accordingly?
  18. Overall, section 3.1., is quite superficial, and the authors did not provide any detailed information on the identified genes. Please check this article https://doi.org/10.3390/ijms22084281 and improve your results accordingly. The authors must provide the sequences of all the genes in a supplementary file.
  19. Section 3.2., lacks original data supporting their results. The authors must provide the detailed results in supplementary tables. Please follow the above-mentioned article to improve the presentation of your work. Please do not simply write what you did, and the authors must explain their results in-depth, like there is no information for the number of intron/exons, etc.
  20. I am stopping here because the rest of the results are not well presented. The result sections must be rewritten. Please follow some published articles and presents your results accordingly.
  21. Line 197, which tissues? Tissues names are not mentioned anywhere in the text. Also, add the meaning of abbreviations in the figure captions.
  22. Line 198-199, explain the expression levels in terms of %/fold change, etc. The same goes for lines 212-220.
  23. Further, to add something new, I suggest predicting the miRNA targeting WRKy genes. This work will not take much time but will add something new to the manuscript.
  24. The discussion must be improved, mainly section 4.3., which is the repetition of the results. The authors must support their findings by arguing some previous reports of the same gene family. Further, avoid repeating results in the methodology.
  25. The English language needs significant improvement. There are several mistakes throughout the text.

Author Response

Dear reviewer,

Thank you for your valuable comments. We have studied the valuable comments from you, the editors carefully, made a significant effort to make the work clearer, and tried our best to revise the manuscript. We highlighted the changes in the revision. The point to point responds to the reviewer’s comments as following:

Comments and Suggestions for Authors

In the study, the authors performed a genome-wide study to identify the WRKY genes in common walnut. This is the first report in walnut, and the WRKY family has been widely reported in several plant species. However, this manuscript was poorly arranged, written, and completed. The authors fail to provide the main supporting results of all the analysis in the supplementary files, making it hard to evaluate the results from the figures completely. Thus, I strongly suggest read some good papers and improve your presentation accordingly. Mainly, below mentioned paper could help you to improve your work. Thus, I cannot support the paper in its present form unless it is thoroughly revised and improved. Some of the major issues are:

  1. Line 17-18, rephrase the sentence like “In total, we identified 103 WRKY genes in common walnut which are clustered into four groups and distributed on 14 chromosomes.

Response: Thank your valuable comments. We revised the sentence as “In total, we identified 103 WRKY genes in the common walnut that are clustered into 4 groups and distributed on 14 chromosomes in the revision at Line 17-19. ”

  1. Line 19, The authors should present the main results of the conserved domain, gene structure, and motif composition in the abstract.

Response: Thank your valuable comments. We added these sentences in the Abstract section as followings:“In total, we identified 103 WRKY genes in the common walnut that are clustered into 4 groups and distributed on 14 chromosomes. The conserved domains all contained a WRKY domain, and motif 2 was observed in most WRKY proteins, suggesting a high degree of conservation and similar functions within each subfamily. However, gene structure was significantly differentiated between different subfamilies. Synteny analysis indicates that there were 56 gene pairs in J. regia and A. thaliana, 76 in J. regia and J. mandshurica, 75 in J. regia and J. microcarpa , 76 in J. regia and P. trichocarpa, and 33 in J. regia and Q. robur, indicating that the WRKY gene family may come from a common ancestor. GO and KEGG enrichment analysis showed that the WRKY gene family was involved in resistance traits and the plant–pathogen interaction pathway. In anthracnose-resistant F26 fruits (AR) and anthracnose-susceptible F423 fruits (AS), transcriptome and qPCR analysis results showed that JrWRKY83, JrWRKY73, and JrWRKY74 were expressed significantly more highly in resistant cultivars, indicating that these three genes may be important contributors to stress resistance in walnut trees. Furthermore, we investigate how these three genes potentially target miRNAs and interact with proteins. JrWRKY73 was target by the miR156 family, including 12 miRNAs; this miRNA family targets WRKY genes to enhance plant defense. JrWRKY73 also interacted with the resistance protein AtMPK6, showing that it may play a crucial role in walnut defense.”

  1. Update the keywords; they should not be similar to the title.

Response: Thank your valuable comments. We revised key words as followings:

“Keywords: Juglans regia; WRKY; resistance genes; miR156; protein interaction.” in the revision at Line 35-36.

  1. Line 34-43, please rephrase the sentences for more clarity. Like, 10 members in cotton, 5 members in ABC, etc. Further, add references after every study.

Response: Thank your valuable comments. We revised these sentences as followings in the revision at Line 40-52:

“ In important crops, WRKY genes have been examined in their related genomes, namely 75 in peanut [3], 174 in soybean [4], 148 in Brassica oleracea [5], 92 in quinoa (Chenopodium quinoa) [6], 126 in Raphanus sativus [7], 79 in potato (Solanum tuberosum) [8], 112 in Gossypium raimondii and 109 in Gossypium arboreum WRKY [9], in cotton (116 in G. raimondii and 102 in Gossypium hirsutum) [10], 89 in rice [11], 63 in Dendrobium officinale [12], and 86 in Barley [13]. In woody plants, WRKY genes have been examined in their related genomes, namely 147 genes in Musa acuminate and 132 in Musa balbisiana [14], 58 in castor bean [2], 54 in pineapple [15], 56 in tea [16], 51 in C. sinensis [16], 48 in C. clementina, 79 in Citrus unshiu [17], 69 in Dimocarpus longan [18], 58 in moso bamboo [19], 85 in Salix suchowensis [20], 59 in peach [21], 95 in Dendrobium officinale  [22], 71 in Fragaria vesca [23], 53 in Elaeis guineensis [24], 71 in sesame [25], and 53 in Caragana intermedia [26]”

  1. Line 47, 50, define SA, JA, MeJA abbreviations. Also, check the entire text for undefined abbreviations like line 66.

Response: Thank your valuable comments. We added the full name of salt (SA) and jasmonate (JA) treatment; methyl jasmonate (MeJA) treatments; phenylalanine ammonialyase (PAL), fatty acid desaturase (FAD), heat stress transcription factors (HSFs), nascent polypeptide-associated complex protein (NAC), and repressor of GAI; Gibberellic acid-insensitive of RGA, and Scarecrow SCR (GRAS); anthracnose-resistant F26 (AR); anthracnose-susceptible F423 (AS) in the revison.

  1. Line 72, uploaded?? What?? Would you please check and correct the statement?

Response: Thank your valuable comments. We deleted the word “uploaded” and added the reference number [41] in the revision at Line 85.

  1. Line 77, 1e10-10, correct it.

Response: Thank your valuable comments. We changed “1e10-10” to “1e-10” in the revision at Line 89.

  1. Line 88, fix the spacing error.

Response: Thank your valuable comments. We fix the spacing error in the revision at Line 99.

  1. In the methodology, the authors use “WE” several times. Please avoid using we, our, etc.

Response: Thank your valuable comments. We changed these describes as following:

“We aligned the complete common protein sequence of walnut WRKY using by MEGA7.0 (State College, PA, USA) software with the default parameters [41]”. to “The complete WRKR protein sequence of walnut was aligned by MEGA7.0 (State College, PA, USA) software with default parameters [45].” at Line 103-104.

“We performed the identification of potential pairs of homologous genes across multiple genomes (E<1×10-5, top 5 matches) using BLASTP (cite). We used homologous gene pairs to identify syntenic chains through MCScanX [45]. We detected duplicate gene pairs by using MCScanX, which included whole-genome duplication (WGD), tandem duplication, segmental, and other types of gene pairs [45].” to “BLASTP was used toan identification of potential pairs of homologous genes across multiple genomes (E<1×10-5, top 5 matches) [42]. The homologous gene pairs were used to identify syntenic chains through MCScanX [50]. The detected duplicate gene pairs were detected byusing MCScanX, which included whole-genome duplication (WGD), tandem duplication, segmental, and other types of gene pairs [50].” at Line 120-124.

“We used Cufflinks [48] with default parameters to quantify these gene expression lev-els based on fragments per million base readings per million reading mappings (FPKM) values and expressed using Hemi 1.0 software with default parameters results [49].” to  “Cufflinks was used to [53] to quantify these gene expression levels based on fragments per million base readings per million mapped read (FPKM) values with default pa-rameters, and expression level was calculated using Hemi 1.0 software with default parameters [54].” at Line 130-134.

 “we download the transcriptome data from NCBI” to “17 transcriptome data were downloaded” at Line 126-128.

  1. Line 94, it should be GSDs. Would you please rephrase the exact database name?

Response: Thank your valuable comments. We added the exact database name:

“Protein structure information was predicted by an online web server of a conserved domain (https://www.ncbi.nlm.nih.gov/Structure/bwrpsb/bwrpsb.cgi) (CDD-search) [49].” at Line 117-118.

  1. Line 95-96, please rephrase the sentences for more clarity.

Response: Thank your valuable comments. We rewrite these sentences as followings in the revision at Line 111-118:

“Feature coordinates (exon–intron boundaries) were extracted from the GFF3

annotation files of walnut. The exon–intron structure was illustrated by TBtools [47]. Identification of patterns using various pattern alignments with the default pattern-initiated (MEME) program parameters was conducted with the maximum number of patterns set to 20, and the optimum pattern width was set to 15-20 [48]. Protein structure information was predicted by an online web server of a conserved domain (https://www.ncbi.nlm.nih.gov/Structure/bwrpsb/bwrpsb.cgi) (CDD-search) [49].”

  1. Line 102, cite??? What?

Response: Thank your valuable comments. We added the reference for BLASTP in the revision at Line 121.

  1. Line 108, add the accession number or URL which is mentioned on line 110.

Response: Thank your valuable comments. We revised these sentences in the revision at line 126-129 as followings:

“ To analyze the expression levels of WRKY in common walnut, 17 transcriptome data were downloaded, including a total of 17 tissues from PJ Martínez-García et al. et al., 2016 (PRJNA291087) [51], 10 anthracnose-resistant F26 fruits from the B26 clone (AR), and 10 anthracnose-susceptible F423 fruits from the 4–23 clone (AS)  of walnut [52] (PRJNA612972) (Table S1).”

  1. Line 114, change genes to Genes.

Response: Thank your valuable comments. We change “Genes” to “genes” in the revision at Line 135.

  1. Line 118, mention the tissues/organs names.

Response: Thank your valuable comments. We added the tissues full name in the revision at Line 136-140 as following:

“We then verified the JrWRKY transcript factors expression profiles in common walnut by quantitative real-time PCR (qRT-PCR) reactions in different tissues (immature fruit, pistillate flower, mature pistillate flower, embryo, somatic embryo, vegetative bud, callus exterior, catkins, hull cortex, immature hull, young hull, mature hull, hull peel, immature leaves, young leaf, mature leaves, root)”

  1. Line 123, fix the superscript issue.

Response: Thank your valuable comments. We revised it at Line 145.

  1. Line 126-129, Table S3 should be included in the main text. Further, more information on the physico-chemical properties of molecular weight, isoelectric points, etc., must be added to this table. Also, explain the method for physioco-chemical properties in the methodology section. Would you please check some recently published article and improve this part accordingly?
  2. Overall, section 3.1., is quite superficial, and the authors did not provide any detailed information on the identified genes. Please check this article https://doi.org/10.3390/ijms22084281 and improve your results accordingly. The authors must provide the sequences of all the genes in a supplementary file.

Response: Thank your valuable comments. We move Table S3 as Table 1 in the main text.

We added the information on the physicochemical properties of subcellular location, molecular weight, isoelectric points of WRKY gene family in walnut as Table 1.

We improved the describes of section 3.1 at Line 160 to 168 as followings:

“3.1. Identification and classification of WRKY genes

In this study, we detected a total of 103 WRKY genes in common walnut (Figure 1; Table 1). Based on the similar domain in which walnut WRKY genes were identified, the WRKY genes were classified into four groups, and the fourth group had the largest number of genes, including fifty-one members. These walnut WRKY proteins ranged in length from 281 to 760 amino acids, with a molecular weight from 12.02 kDa to 56.39 kDa and isoelectric points ranging from 4.97 to 9.72. Subcellular localization analysis indicated that all 103 walnut WRKY genes were localized in the nucleus. (Figure 1; Table 1; Table S3).”

We added the sequences of all the genes in a Supplementary Table 3.

  1. Section 3.2., lacks original data supporting their results. The authors must provide the detailed results in supplementary tables. Please follow the above-mentioned article to improve the presentation of your work. Please do not simply write what you did, and the authors must explain their results in-depth, like there is no information for the number of intron/exons, etc.

Response: Thank your valuable comments. We added all the original data supporting results in Supplementary Materials section.

“Table S3. The sequence of WRKY gene family. Table S4. The gene structure information of WRKY genes; Table S5. The collinearity in J.regia and their related information; Table S7. The collinearity in J. regia, A. thaliana, J. mandshurica, J. microcarpa, P. trichocarpa, and Q. robur and their related information; TableS8. The KEGG enrichment analysis of WRKY genes in J. regia; Table S9. The WRKY genes expression level in different tissues of J. regia; Table S10. The WRKY genes expression level in AR and AS fruits; Table S11. The qPCR data of three genes; Table S12. The predicted miRNA sequence target WRKY genes; Table S13. The target WRKY genes sequence were targeted by miRNAs; Table S14. The predicted miRNA and target genes interaction network; Table S15. The regulatory network relationships between putative miRNAs and JrWRKY73

We added some describes at section 3.2 in the revision at Line 180-197 as followings:

“According to the phylogenetic tree and motif composition, these gene families were divided into seven subfamilies (Figure 1). According to the gene structure and conserved motif distribution, WRKY genes showed diverse sequence structures (Figure 2; Table S4). In the present study, 15 conserved motifs were detected in WRKY proteins, and motif 2 was observed in most proteins as a subgroup that contained the most motifs (7), while the fewest motifs were found in subgroup 6, which contained only 3 motifs (Figure 2a, b). All WRKY genes contained at least one WRKY conserved domain, but subgroup 3 WRKY genes contained 2 WRKY domains; only two genes (JrWRKY76 and JrWRKY77) contained WRKY and CCCC73 domains (Figure 2c). In addition, WRKY genes were diverse in terms of gene structure, where various intron–exon numbers were observed (Figure 2d), which proved the validity of the phylogenetic tree and motif composition. The structure of WRKY genes in common walnut has different exon–intron organizations between subfamilies (Figure 4; Table S4). Intron numbers 2 to 6 were found in all WRKY genes (Figure 2d). Subgroup 1 contained 5 to 6 exons; subgroup 2 contained 3 to 4 exons; subgroup 3 contained 5 to 6 exons; subgroup 4 contained 4 to 5 exons; subgroup 5 contained 2 exons; subgroup 6 contained 3 exons; and subgroup 7 contained 3 to 4 exons (Figure 2d). ”

  1. I am stopping here because the rest of the results are not well presented. The result sections must be rewritten. Please follow some published articles and presents your results accordingly.

Response: Thank your valuable comments. We revised the describes of 3.3. Chromosome distribution and synteny analysis of WRKY genes; 3.4. GO and KEGG enrichment analysis of WRKY gene family in walnut; 3.5. Three genes (JrWRKY83, JrWRKY73 and JrWRKY74) may involve in resistance traits of walnut based on transcriptome data and qPCR; 3.6. MicroRNA targeting and WRKY target genes.

  1. Line 197, which tissues? Tissues names are not mentioned anywhere in the text. Also, add the meaning of abbreviations in the figure captions.

Response: Thank your valuable comments. We added the meaning of abbreviations for Figure8a at Line 272-276.

“IF6: immature fruit; FL3: pistillate flower; FL6: mature pistillate flower; EM8: embryo; SE7: somatic embryo; VB5: vegetative bud; CE5: callus exterior; CK3: catkins; HC2: hull cortex; HU3: immature hull; HL1:immature hull; HL6: young hull ; HP3: hull peel; LY2: immature leaf; LY7: young leaf; LE5: mature leaves; RT6: root. (b) Mean box plot of WRKY members between anthracnose-resistant F26 fruits (AR) and an-thracnose-susceptible F423 fruits (AS); each black circle represents each sample.”

  1. Line 198-199, explain the expression levels in terms of %/fold change, etc. The same goes for lines 212-220.

Response: Thank your valuable comments. We added the fold change in the revision at Line 259-270 as followings:

“particularly JrWRKY93, JrWRKY94, JrWRKY83, JrWRKY73, and JrWRKY74 (Figure 8a; Table S9). A mean box plot shows that there were three genes (JrWRKY83 :546-fold, JrWRKY73: 307-fold, and JrWRKY74: 1920-fold) highly expressed in anthracnose-resistant F26 fruits”

“ Based on our real-time PCR results, we observed that JrWRKY83 (23-fold), JrWRKY73 (10-fold), and JrWRKY74 (11-fold) were highly expressed in resistance traits compared to non-resistance traits, including leaf and fruit (Figure 8e-f; Table S11).”

  1. Further, to add something new, I suggest predicting the miRNA targeting WRKy genes. This work will not take much time but will add something new to the manuscript.

Response: Thank your valuable comments. We added the miRNA Predicted in walnut WRKY family genes and the interaction network of JrWRKY proteins as followings:

In the methods section: 2.7. miRNA Predicted in walnut WRKY family genes and the interaction network of JrWRKY proteins.

“All of the genome sequences of the common walnut WRKY family genes were submitted as candidate genes to predict potential miRNAs by searching against the available walnut reference of miRNA sequences using the psRNATarget Server with default parameters [56]. We visualized the interactions between the predicted miRNAs and the corresponding target walnut WRKY genes using Cytoscape software with default parameters [57]. Persian walnut WRKY protein matched a homologous Arabidopsis WRKY protein in the BLASTP program with an E value of 1e-05 [58]. Regarding the Arabidopsis WRKY proteins that represented the walnut WRKY proteins, 102 were uploaded to the STRING website to predict protein interactions (https://string-db.org/) [59].”

In the results section: “ 3.6. MicroRNA targeting and WRKY interaction network”

“To understand the underlying regulatory mechanism of miRNAs involved in the regulation of WRKYs, we identified 206 putative miRNAs targeting 45 common walnut WRKY genes (Figure9a; Table S12-14). The most target genes were JrWRKY65 and JrWRKY67, containing 197 miRNAs, while the least targeted gene was JrWRKY55, containing 62 miRNAs (Figure9a; Table S12-14). Based on transcriptome profile and qPCR results, we selected JrWRKY73 and the related 85 miRNAs to construct a relationship network using Cytoscape software (Figure 9a; Table S15). Of these 85 miRNAs, we found that the miRNA family with the closest relationship was JrWRKY73, which was targeted by the Jre-miR156 family, including 12 miRNAs (Jre-miR156a, Jre-miR156b, Jre-miR156c, Jre-miR156d, Jre-miR156e, Jre-miR156f, Jre-miR156g, Jre-miR156h, Jre-miR156i, Jre-miR156j, Jre-miR156k, Jre-miR156l) (Figure 9a; Table S15). Each JrWRKY protein was in close association with at least one WRKY protein from Arabidopsis. Some JrWRKY proteins were closely aligned with the same WRKY protein in Arabidopsis. We downloaded WRKY proteins from Arabidopsis to detect the predicted role of highly expressed genes in the fruits and leaves of AR of Persian walnut. A previous study claimed that these genes regulate the development of fruits and are responsible for stress. Therefore, we detected the interaction relationship between these genes, and the results indicate a strong relationship between JrWRKY73 proteins and AtCYP78A9, AtMPK6, AtMPK10, AtARF19, and AtCYP78A9 (Figurer 9b).”

In the discussion section: 4.3. The function of WRKY gene family

“In recent years, many studies have shown that miRNAs in plants respond primarily to stress by regulating the expression of genes associated with stress [71]. In terms of WRKY, some researchers have reported that Md-miR156ab and Md-miR395 resulted in a significant reduction in MdWRKYN1 and MdWRKY26 expression [72]. HaWRKY6 is a particularly divergent WRKY gene exhibiting a putative target site for the miR396; thus, the possible post-transcriptional regulation of HaWRKY6 by miR396 was investigated [73]. In our study, we found that 12 miRNAs of the miRNA156 family targeted the potentially resistant gene JrWRKY73, and we also reported that Md-miR156ab targeted WRKY transcription factors to influence apple resistance to leaf spot disease [72]. The diverse patterns of microRNA targeting WRKY genes indicate that the networks of microRNA156 and JrWRKY73 may be key regulator networks for the WRKY gene family in common walnut. The results of the interaction indicate a strong relationship between JrWRKY73 proteins and AtCYP78A9 [74], AtMPK6 [75], AtMPK10 [76], and AtARF19[77]. AtCYP78A9 induces large and seedless fruit in Arabidopsis, indicating that JrWRKY73 may participate in the development of walnut fruits [74]. JrWRKY73 interacts with AtMPK10 and AtMPK13, while AtMKK6 and AtMPK4 activate AtMPK13 and interact with AtMPK12 in yeast cells, indicating that they may have the same function [76]. JrWRKY73 interacts with the activator of a cholera toxin known as AtARF19, indicating that JrWRKY73 may have the same function [77]. JrWRKY73 showed a higher expression level in AS fruits when induced by C. gloeosporioides stress, which was consistent with previous studies reporting that ATMPK6 was involved in distinct signal transduction pathways responding to these environmental stresses [75]. These protein interactions showed that JrWRKY73 may play a key role in fruit development, yeast cells, activation of a cholera toxin, and resistance in walnut. However, when combined with the expression profile, miRNA-targeted network, and protein interacted network, the results showed that JrWRKY73 played critical role in walnut defense. Moreover, these findings could lay a theoretical foundation for the functional study of JrWRKYs and the further construction of common walnut resistance regulation networks.”

  1. The discussion must be improved, mainly section 4.3., which is the repetition of the results. The authors must support their findings by arguing some previous reports of the same gene family. Further, avoid repeating results in the methodology.

Response: Thank your valuable comments. We revised and improved the section 4.3 as followings:

 “The WRKY gene family is involved in many important biology processes, though the most important are response to abiotic stresses, pathogen defense, senescence, and trichome development [3, 7, 8, 10, 22, 30-31, 63]. There were 20% (4/20) GO biological process pathways and 50% (1/2) KEGG pathways that were enriched in resistance pathways, which is consistent with previous studies claiming that WRKY genes work against abiotic and biotic stresses [30-31]. We were interested in WRKY genes that regulate the development of resistance traits; therefore, we analyzed public transcriptome data for J. regia (Table S1, Figure 8) and discovered that many family members were highly expressed in the hull. The hull is always impacted by stress [69], indicating that WRKYs may be responsible for conferring stress resistance in walnut (Figure 8) [30-31]. With the transcriptome data of AR fruits and AS fruits, we investigated the expression profile between these two fruits; three genes, JrWRKY83, JrWRKY732, and JrWRKY74, were highly expressed in AR fruits, indicating that these three genes increased their expression level when infected by the stress of Colletotrichum gloeosporioides, which predominantly affects walnut anthracnose through C. gloeosporioides can cause leaf scorches or defoliation, as well as fruit gangrene, which is currently the most challenging disease in walnut production [70]. In line with previous studies, we collected leaves and fruits infected by C. gloeosporioides. Regarding our real-time PCR results, JrWRKY83, JrWRKY732, and JrWRKY74 were induced by C. gloeosporioides stress in the leaves and root tissues of walnut cultivars. JrWRKY83, JrWRKY732, and JrWRKY74 were more upregulated in response to C. gloeosporioides stress. In recent years, many studies have shown that miRNAs in plants respond primarily to stress by regulating the expression of genes associated with stress [71]. In terms of WRKY, some researchers have reported that Md-miR156ab and Md-miR395 resulted in a significant reduction in MdWRKYN1 and MdWRKY26 expression [72]. HaWRKY6 is a particularly divergent WRKY gene exhibiting a putative target site for the miR396; thus, the possible post-transcriptional regulation of HaWRKY6 by miR396 was investigated [73]. In our study, we found that 12 miRNAs of the miRNA156 family targeted the potentially resistant gene JrWRKY73, and we also reported that Md-miR156ab targeted WRKY transcription factors to influence apple resistance to leaf spot disease [72]. The diverse patterns of microRNA targeting WRKY genes indicate that the networks of microRNA156 and JrWRKY73 may be key regulator networks for the WRKY gene family in common walnut. The results of the interaction indicate a strong relationship between JrWRKY73 proteins and AtCYP78A9 [74], AtMPK6 [75], AtMPK10 [76], and AtARF19[77]. AtCYP78A9 induces large and seedless fruit in Arabidopsis, indicating that JrWRKY73 may participate in the development of walnut fruits [74]. JrWRKY73 interacts with AtMPK10 and AtMPK13, while AtMKK6 and AtMPK4 activate AtMPK13 and interact with AtMPK12 in yeast cells, indicating that they may have the same function [76]. JrWRKY73 interacts with the activator of a cholera toxin known as AtARF19, indicating that JrWRKY73 may have the same function [77]. JrWRKY73 showed a higher expression level in AS fruits when induced by C. gloeosporioides stress, which was consistent with previous studies reporting that ATMPK6 was involved in distinct signal transduction pathways responding to these environmental stresses [75]. These protein interactions showed that JrWRKY73 may play a key role in fruit development, yeast cells, activation of a cholera toxin, and resistance in walnut. However, when combined with the expression profile, miRNA-targeted network, and protein interacted network, the results showed that JrWRKY73 played critical role in walnut defense. Moreover, these findings could lay a theoretical foundation for the functional study of JrWRKYs and the further construction of common walnut resistance regulation networks.”

  1. The English language needs significant improvement. There are several mistakes throughout the text.

Response: Thank your valuable comments. We submit our manuscript to MDPI to improve English language.

Thank you for your consideration our manuscript of “Genome-wide identification and transcriptional expression profiles of transcription factor WRKY in common walnut (Juglans regia L.)" publish on Genes.

We are looking forward to hearing from you.

Yours sincerely,

Shuoxin Zhang

Reviewer 2 Report

The manuscript by Hao et al represent a  study of WRKY transcription factors in common walnut, their expression in different tissues and identification of three WRKY transcription factors which are differentially regulated in anthracnose-sensitive and resistant cultivars.

WRKY transcription factors play an important role in plant development and stress responses, therefore the study is quite interesting and provide new information for this gene family in common walnut.

I have several comments on the manuscript:

  1. Figure 2: it is impossible to read the figure in the current format, could it be larger?
  2. The authors say that there is no WRKY genes located on chromosomes 6 and 15 (page 5). However, Figures 3 and 4 indicate WRKY genes in these chromosomes. Also, chromosome 10 contains 10 WRKY genes according to the figure 3 and 4, but not 9 as it is said in the text (page 5).
  3. Figure 8 (b): the dots are almost invisible. Also the Y axis labels are not indicated on the panels (a) and (b). The tissue names are given in abbreviation (x-axis on panel a), what are these abbreviations?
  4. Figure 9 represents qPCR analysis of three WRKY genes in sensitive and resistant cultivars. The primer sequences are given in Supplementary table S2, however, different nomenclature is used in the figure and in the table. Please use the same nomenclature for the data.

Overall, the study is systematic and provide new data on WRKY family in walnut; however, data presentation can be improved.

Author Response

Dear reviewer,

Thank you for your valuable comments. We have studied the valuable comments from you, the editors carefully, made a significant effort to make the work clearer, and tried our best to revise the manuscript. We highlighted the changes in the revision. The point to point responds to the reviewer’s comments as following:

Comments and Suggestions for Authors

The manuscript by Hao et al represent a study of WRKY transcription factors in common walnut, their expression in different tissues and identification of three WRKY transcription factors which are differentially regulated in anthracnose-sensitive and resistant cultivars.

WRKY transcription factors play an important role in plant development and stress responses, therefore the study is quite interesting and provide new information for this gene family in common walnut.

I have several comments on the manuscript:

  1. Figure 2: it is impossible to read the figure in the current format, could it be larger?

Response: Thank your valuable comments. We revised it in the revision.

The authors say that there is no WRKY genes located on chromosomes 6 and 15 (page 5). However, Figures 3 and 4 indicate WRKY genes in these chromosomes. Also, chromosome 10 contains 10 WRKY genes according to the figure 3 and 4, but not 9 as it is said in the text (page 5).

Response: Thank your valuable comments. We revised these describes in the revision at Line 203-206 as followings:

“All WRKY genes could be mapped onto 12 chromosomes of walnut. Chromosome 10 contained the highest number of WRKY genes (10), whereas the fewest WRKY genes were located on chromosome 5 (1) (Figure 3).”

  1. Figure 8 (b): the dots are almost invisible. Also, the Y axis labels are not indicated on the panels (a) and (b). The tissue names are given in abbreviation (x-axis on panel a), what are these abbreviations?

Response: Thank your valuable comments. We revised these describes as followings:

“Figure 8. The expression profile of WRKY gene family in walnut: (a) the highly expressed genes among different tissues of walnut; IF6: immature fruit; FL3: pistillate flower; FL6: mature pistillate flower; EM8: embryo; SE7: somatic embryo; VB5: vegetative bud; CE5: callus exterior; CK3: catkins; HC2: hull cortex; HU3: immature hull; HL1:immature hull; HL6: young hull ; HP3: hull peel; LY2: immature leaf; LY7: young leaf; LE5: mature leaves; RT6: root. (b) Mean box plot of WRKY members between anthracnose-resistant F26 fruits (AR) and anthracnose-susceptible F423 fruits (AS); each black circle represents each sample. (c-d) The morphology of walnut leaf (resistance (Cultivar Xiangling) and non-resistance (Cultivar Shanhe5)) and fruit (re-sistance (Cultivar Xiangling) and non-resistance (Cultivar Shanhe5)). (e-f) Relative expression levels of WRKY genes in walnut leaf (resistant cultivar Xiangling and non-resistant cultivar Shanhe5, and fruit resistant cultivar Xiangling and non-resistant cultivar Shanhe5).”

  1. Figure 9 represents qPCR analysis of three WRKY genes in sensitive and resistant cultivars. The primer sequences are given in Supplementary table S2, however, different nomenclature is used in the figure and in the table. Please use the same nomenclature for the data.

Response: Thank your valuable comments. We revised the gene name of Table S2.

  1. Overall, the study is systematic and provide new data on WRKY family in walnut; however, data presentation can be improved.

Response: Thank your valuable comments.

We added the miRNA Predicted in walnut WRKY family genes and the interaction network of JrWRKY proteins as followings:

In the methods section: 2.7. miRNA Predicted in walnut WRKY family genes and the interaction network of JrWRKY proteins.

“All of the genome sequences of the common walnut WRKY family genes were submitted as candidate genes to predict potential miRNAs by searching against the available walnut reference of miRNA sequences using the psRNATarget Server with default parameters [56]. We visualized the interactions between the predicted miRNAs and the corresponding target walnut WRKY genes using Cytoscape software with default parameters [57]. Persian walnut WRKY protein matched a homologous Arabidopsis WRKY protein in the BLASTP program with an E value of 1e-05 [58]. Regarding the Arabidopsis WRKY proteins that represented the walnut WRKY proteins, 102 were uploaded to the STRING website to predict protein interactions (https://string-db.org/) [59].”

In the results section: “ 3.6. MicroRNA targeting and WRKY interaction network”

“To understand the underlying regulatory mechanism of miRNAs involved in the regulation of WRKYs, we identified 206 putative miRNAs targeting 45 common walnut WRKY genes (Figure9a; Table S12-14). The most target genes were JrWRKY65 and JrWRKY67, containing 197 miRNAs, while the least targeted gene was JrWRKY55, containing 62 miRNAs (Figure9a; Table S12-14). Based on transcriptome profile and qPCR results, we selected JrWRKY73 and the related 85 miRNAs to construct a relationship network using Cytoscape software (Figure 9a; Table S15). Of these 85 miRNAs, we found that the miRNA family with the closest relationship was JrWRKY73, which was targeted by the Jre-miR156 family, including 12 miRNAs (Jre-miR156a, Jre-miR156b, Jre-miR156c, Jre-miR156d, Jre-miR156e, Jre-miR156f, Jre-miR156g, Jre-miR156h, Jre-miR156i, Jre-miR156j, Jre-miR156k, Jre-miR156l) (Figure 9a; Table S15). Each JrWRKY protein was in close association with at least one WRKY protein from Arabidopsis. Some JrWRKY proteins were closely aligned with the same WRKY protein in Arabidopsis. We downloaded WRKY proteins from Arabidopsis to detect the predicted role of highly expressed genes in the fruits and leaves of AR of Persian walnut. A previous study claimed that these genes regulate the development of fruits and are responsible for stress. Therefore, we detected the interaction relationship between these genes, and the results indicate a strong relationship between JrWRKY73 proteins and AtCYP78A9, AtMPK6, AtMPK10, AtARF19, and AtCYP78A9 (Figurer 9b).”

In the discussion section: 4.3. The function of WRKY gene family

“In recent years, many studies have shown that miRNAs in plants respond primarily to stress by regulating the expression of genes associated with stress [71]. In terms of WRKY, some researchers have reported that Md-miR156ab and Md-miR395 resulted in a significant reduction in MdWRKYN1 and MdWRKY26 expression [72]. HaWRKY6 is a particularly divergent WRKY gene exhibiting a putative target site for the miR396; thus, the possible post-transcriptional regulation of HaWRKY6 by miR396 was investigated [73]. In our study, we found that 12 miRNAs of the miRNA156 family targeted the potentially resistant gene JrWRKY73, and we also reported that Md-miR156ab targeted WRKY transcription factors to influence apple resistance to leaf spot disease [72]. The diverse patterns of microRNA targeting WRKY genes indicate that the networks of microRNA156 and JrWRKY73 may be key regulator networks for the WRKY gene family in common walnut. The results of the interaction indicate a strong relationship between JrWRKY73 proteins and AtCYP78A9 [74], AtMPK6 [75], AtMPK10 [76], and AtARF19[77]. AtCYP78A9 induces large and seedless fruit in Arabidopsis, indicating that JrWRKY73 may participate in the development of walnut fruits [74]. JrWRKY73 interacts with AtMPK10 and AtMPK13, while AtMKK6 and AtMPK4 activate AtMPK13 and interact with AtMPK12 in yeast cells, indicating that they may have the same function [76]. JrWRKY73 interacts with the activator of a cholera toxin known as AtARF19, indicating that JrWRKY73 may have the same function [77]. JrWRKY73 showed a higher expression level in AS fruits when induced by C. gloeosporioides stress, which was consistent with previous studies reporting that ATMPK6 was involved in distinct signal transduction pathways responding to these environmental stresses [75]. These protein interactions showed that JrWRKY73 may play a key role in fruit development, yeast cells, activation of a cholera toxin, and resistance in walnut. However, when combined with the expression profile, miRNA-targeted network, and protein interacted network, the results showed that JrWRKY73 played critical role in walnut defense. Moreover, these findings could lay a theoretical foundation for the functional study of JrWRKYs and the further construction of common walnut resistance regulation networks.”

We added all the original data supporting results in Supplementary Materials section.

“Table S3. The sequence of WRKY gene family. Table S4. The gene structure information of WRKY genes; Table S5. The collinearity in J.regia and their related information; Table S7. The collinearity in J. regia, A. thaliana, J. mandshurica, J. microcarpa, P. trichocarpa, and Q. robur and their related information; TableS8. The KEGG enrichment analysis of WRKY genes in J. regia; Table S9. The WRKY genes expression level in different tissues of J. regia; Table S10. The WRKY genes expression level in AR and AS fruits; Table S11. The qPCR data of three genes; Table S12. The predicted miRNA sequence target WRKY genes; Table S13. The target WRKY genes sequence were targeted by miRNAs; Table S14. The predicted miRNA and target genes interaction network; Table S15. The regulatory network relationships between putative miRNAs and JrWRKY73

We added some describes at section 3.2 in the revision at Line 180-197 as followings:

“According to the phylogenetic tree and motif composition, these gene families were divided into seven subfamilies (Figure 1). According to the gene structure and conserved motif distribution, WRKY genes showed diverse sequence structures (Figure 2; Table S4). In the present study, 15 conserved motifs were detected in WRKY proteins, and motif 2 was observed in most proteins as a subgroup that contained the most motifs (7), while the fewest motifs were found in subgroup 6, which contained only 3 motifs (Figure 2a, b). All WRKY genes contained at least one WRKY conserved domain, but subgroup 3 WRKY genes contained 2 WRKY domains; only two genes (JrWRKY76 and JrWRKY77) contained WRKY and CCCC73 domains (Figure 2c). In addition, WRKY genes were diverse in terms of gene structure, where various intron–exon numbers were observed (Figure 2d), which proved the validity of the phylogenetic tree and motif composition. The structure of WRKY genes in common walnut has different exon–intron organizations between subfamilies (Figure 4; Table S4). Intron numbers 2 to 6 were found in all WRKY genes (Figure 2d). Subgroup 1 contained 5 to 6 exons; subgroup 2 contained 3 to 4 exons; subgroup 3 contained 5 to 6 exons; subgroup 4 contained 4 to 5 exons; subgroup 5 contained 2 exons; subgroup 6 contained 3 exons; and subgroup 7 contained 3 to 4 exons (Figure 2d). ”

We submit our manuscript to MDPI to improve English language.

Thank you for your consideration our manuscript of “Genome-wide identification and transcriptional expression profiles of transcription factor WRKY in common walnut (Juglans regia L.)" publish on Genes.

We are looking forward to hearing from you.

Yours sincerely,

Shuoxin Zhang

Round 2

Reviewer 1 Report

This is a much-improved version, and the authors had addressed all the comments satisfactorily. However, some minor corrections should be made .

  1. Line 40, namely??? What? Please correct it.
  2. Line 191, in the table, the values are according to Da instead of kDa. Would you please modify the values in the table or change the unit in the footnote? Further, remove dot after nucleus.
  3. The gene names are not consistent throughout the text. Please carefully check the entire text and make sure all genes are italics, not the protein name.
  4. At the end of the conclusion, please add future directions.
  5. The entire text should be proof check for spacing issues.

Author Response

Dear reviewer,

Thank you for your valuable comments. We have studied the valuable comments from you and the editors carefully, made a significant effort to make the work clearer, and tried our best to revise the manuscript. We highlighted the changes in the revision. The point to point responds to the reviewer’s comments as following:

This is a much-improved version, and the authors had addressed all the comments satisfactorily. However, some minor corrections should be made.

1.Line 40, namely??? What? Please correct it.

Response: Thank your valuable comments. We changed words “namely” to “including” in the revision at Line 40.

2.Line 191, in the table, the values are according to Da instead of kDa. Would you please modify the values in the table or change the unit in the footnote? Further, remove dot after nucleus.

Response: Thank your valuable comments. We changed words “kDa” to “Da” in the revision at Line 163 and footnote. We remove dot after nucleus in the revision.

3.The gene names are not consistent throughout the text. Please carefully check the entire text and make sure all genes are italics, not the protein name.

Response: Thank your valuable comments. We carefully check the entire text and make sure all genes are italics at Line 20, 33, 83, 85, 89, 90, 93, 101, 107, 151, 152, 163, 196, 288-290, 294-295, 298, 290, 377-378, 410-411 in the revision.

4.At the end of the conclusion, please add future directions.

Response: Thank your valuable comments. We added future directions in the conclusion section.

5.The entire text should be proof check for spacing issues.

Response: Thank your valuable comments. We checked spacing issues in the entire text.

Thank you for your consideration our manuscript of “Genome-wide identification and transcriptional expression profiles of transcription factor WRKY in common walnut (Juglans regia L.)" publish on Genes.

We are looking forward to hearing from you.

Yours sincerely,

Shuoxin Zhang
